# Evolution of limb development in cephalopod mollusks

**Oscar A Tarazona[1,2], Davys H Lopez[1], Leslie A Slota[2], Martin J Cohn[1,2]***

[1]Department of Molecular Genetics and Microbiology, University of Florida, Gainesville, United States; [2]Department of Biology, UF Genetics Institute, University of Florida, Gainesville, United States

**Abstract** Cephalopod mollusks evolved numerous anatomical novelties, including arms and tentacles, but little is known about the developmental mechanisms underlying cephalopod limb evolution. Here we show that all three axes of cuttlefish limbs are patterned by the same signaling networks that act in vertebrates and arthropods, although they evolved limbs independently. In cuttlefish limb buds, *Hedgehog* is expressed anteriorly. Posterior transplantation of *Hedgehog*-expressing cells induced mirror-image limb duplications. Bmp and Wnt signals, which establish dorsoventral polarity in vertebrate and arthropod limbs, are similarly polarized in cuttlefish. Inhibition of Bmp2/4 dorsally caused ectopic expression of *Notum*, which marks the ventral sucker field, and ectopic sucker development. Cuttlefish also show proximodistal regionalization of *Hth*, *Exd*, *Dll*, *Dac*, *Sp8/9*, and *Wnt* expression, which delineates arm and tentacle sucker fields. These results suggest that cephalopod limbs evolved by parallel activation of a genetic program for appendage development that was present in the bilaterian common ancestor.

DOI: https://doi.org/10.7554/eLife.43828.001

## Introduction

Animal appendages have widely varying morphologies and perform a multitude of functions, including locomotion, feeding, and reproduction (*Nielsen, 2012*; *Ruppert et al., 2004*). Limbs evolved on multiple occasions, and the absence of shared ontogenetic or morphological precursors of appendages in many animal lineages is consistent with their independent origins (*Minelli, 2003*; *Pueyo and Couso, 2005*; *Shubin et al., 1997*). This has led to the view that appendages in different clades of Bilateria are non-homologous morphological innovations that arose by convergent evolution (*Nielsen, 2012*; *Ruppert et al., 2004*). However, despite more than 500 million years of divergence, the independently evolved limbs of arthropods and vertebrates share developmental genetic similarities (*Pueyo and Couso, 2005*; *Shubin et al., 1997*; *Tabin et al., 1999*).

These discoveries led to debate over whether the genetic program for appendage development evolved in the common ancestor of all bilaterians in the early Cambrian, or whether arthropod and vertebrate appendages have undergone rampant convergence of developmental programs (*Minelli, 2000*; *Minelli, 2003*; *Panganiban et al., 1997*; *Pueyo and Couso, 2005*; *Shubin et al., 1997*; *Tabin et al., 1999*). A major obstacle to resolving this question is that the evidence of a conserved program derives almost exclusively from Ecdysozoa and Deuterostomia (*Pueyo and Couso, 2005*; *Shubin et al., 1997*), and little is known about molecular mechanisms of limb development in Spiralia, the third major superphylum of Bilateria (*Grimmel et al., 2016*; *Prpic, 2008*; *Winchell and Jacobs, 2013*; *Winchell et al., 2010*).

Within spiralians, the phylum Mollusca is the largest lineage, displaying a rich diversity of body plans (*Figure 1A*) dating back to the Cambrian explosion (*Ruppert et al., 2004*; *Smith et al., 2011*). The evolution of arms and tentacles in cephalopod mollusks contributed to the successful adaptive radiation of these agile marine predators (*Kröger et al., 2011*; *Ruppert et al., 2004*). Cephalopod

*For correspondence:
mjcohn@ufl.edu

**Competing interests:** The authors declare that no competing interests exist.

**eLife digest** Legs, wings, flippers and tentacles are just some examples of the diverse variety of animal limbs. Despite striking differences in form and function, all limbs develop in embryos using similar fundamental processes, like producing an outgrowth from the body and placing structures such as fingers, feathers, or suckers at appropriate positions. Animals have solved this problem multiple times during the history of life on Earth, in that limbed animals have arisen from limbless ancestors on many separate occasions. It is not clear, however, whether the same genetic instructions shape the developing limbs of all species.

Species that have limbs fall under three main groups of animals: arthropods, such as insects and crustaceans; vertebrates, like amphibians, reptiles and mammals; and a specialized group of mollusks known as cephalopods, which includes squid, cuttlefish and octopuses. It has been over two decades since the discovery that the limbs of vertebrates and insects develop using a similar molecular recipe, but the mechanisms responsible for the limbs of cephalopods had not been determined.

Tarazona et al. have now established that the genetic mechanisms that control how cuttlefish limbs develop are the same as those used by the limbs of vertebrates and insects. These mechanisms are also applied for similar purposes in each animal group. Notably, a signaling pathway called hedgehog, which controls the number of fingers that develop on a hand, also dictates the number of suckers on a cuttlefish arm. This may mean that an ancient system for creating limbs emerged over 500 million years ago in the earliest animals with bilateral symmetry (i. e., animals with mirror image halves), and activating this ancient genetic program resulted in the evolution of limbs in different animal lineages.

The extent of the genetic similarities between cuttlefish, mammals and insects suggests that this mechanism is likely to provide instructions about where cells position themselves in the developing limb. The next step is to examine how these common systems are interpreted differently to give arms, legs, wings and other limb forms.

DOI: https://doi.org/10.7554/eLife.43828.002

limbs are highly muscular appendages that bear cup-shaped suckers on their ventral sides. Arms are short and have suckers along the entire ventral surface (*Figure 1B and C*), whereas tentacles are longer, retractable appendages with suckers restricted to a distal pad (*Figure 1D and E*). Tentacles are thought to be specialized serial homologs of the arms (*Arnold, 1965*; *Lemaire, 1970*; *Shigeno et al., 2008*) and are present in decapods (squid and cuttlefish) but absent in nautilids and octopods. Limbs likely evolved de novo in cephalopods (*Figure 1A*), since no homologous precursor structures have been identified in any other mollusk lineages (*Lee et al., 2003*; *Shigeno et al., 2008*). To test the hypothesis that cephalopod limbs evolved by recruitment of an ancient gene regulatory network for appendage development that is conserved across Bilateria, we investigated arm and tentacle development in embryos of the cuttlefish, *Sepia officinalis*.

## Results

### Development of arms and tentacles in the cuttlefish (*Sepia officinalis*)

Cuttlefishes are decapod cephalopods that have eight arms and two tentacles (*Figure 1B–E*; *Figure 1—videos 1* and *2*). Fertilized cuttlefish eggs undergo superficial cleavage, and scanning electron microscopy and optical projection tomography show that most embryonic development is restricted to the animal pole (*Figure 1H and I*). The first sign of limb formation is observed at stage 16, when all ten limb primordia (five on each side) can be detected as small swellings around the periphery of a flat-shaped embryo, which lies at the top of the large yolk mass (*Figure 1H and M*). Analysis of the mitotic marker phospho-histone H3 (PHH3) at stage 15 revealed localized clusters of PHH3-positive cells in each of the early limb primordia (*Figure 1F and G*), indicating that initiation of limb outgrowth is caused by localized cell proliferation. Discrete limb buds are observed from stage 17 (*Figure 1I and N*; *Figure 1—video 3*). As the embryo begins to rise-up on the animal pole around stage 19, the limb buds start to elongate along the proximodistal axis (*Figure 1J and O*;

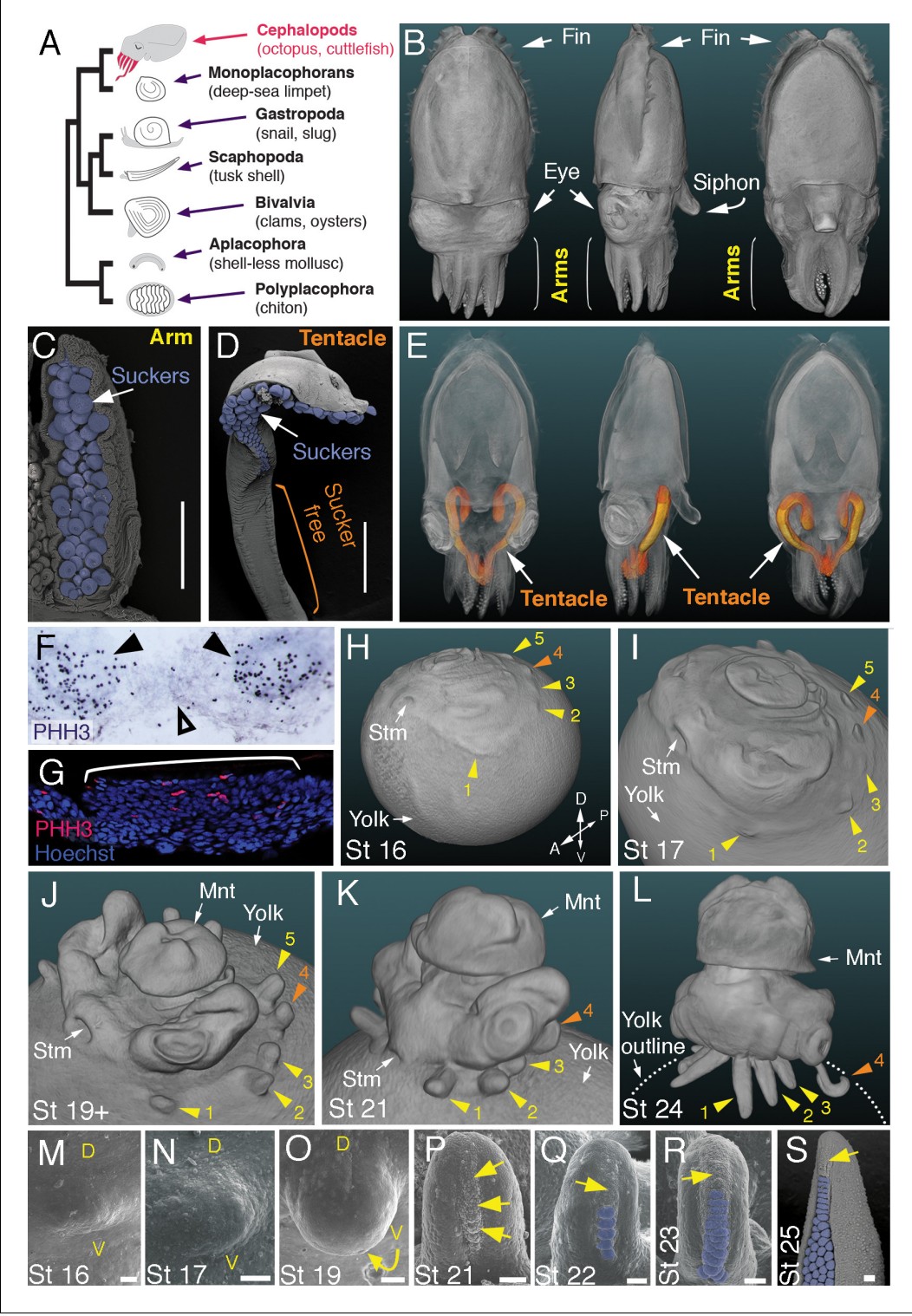

**Figure 1.** Development of arms and tentacles in the cuttlefish, *Sepia officinalis*. (**A**) Phylogenetic relationships of Mollusca based on phylogenomic data (***Smith et al., 2011***) illustrating the unique morphology of the cephalopod body plan compared to other mollusks. (**B**) OPT reconstruction of a cuttlefish hatchling showing positions of the limbs; only arms are visible (see also ***Figure 1—video 1***). (**C** to **D**), SEM of the ventral side of a cuttlefish arm (**C**) and tentacle (**D**). Suckers are pseudocolored blue. Note distal restriction of suckers in tentacle relative to arm. (**E**) OPT reconstruction illustrating the internally retracted tentacles. Specimens are same as in (**B**), but here the tentacles are displayed in orange and the rest of the tissue is rendered translucent (see also ***Figure 1—video 2***).
*Figure 1 continued on next page*

*Figure 1 continued*

(**F and G**) Phospho-histone H3 (PHH3) immunostaining at stage 15 shows localized clusters of proliferating cells at the onset of limb development (black arrowheads) but little proliferation in the interlimb region (open arrowhead). (**F**) PHH3 detection by colorimetric reaction with DAB in a whole mount. (**G**) PHH3 immunofluorescence (red) on limb cryosection (white bracket). Cell nuclei (blue) are labeled by Hoechst staining. (**H-L**) OPT reconstructions of cuttlefish embryos at stages 16 to 24. Cuttlefishes have five bilaterally symmetric limb pairs (ten limbs; eighth arms and two tentacles). Numbered arrowheads mark all five limbs/limb buds on the left side of each embryo. The left tentacle differentiates from position number four (orange arrowhead), whereas arms form from limb buds at the other positions (yellow arrowheads). See also *Figure 1—videos 3—5*. A, anterior; P, posterior; D, dorsal; V, ventral; Stm, stomodeum; Mnt, mantle. (**M to O**) SEM during early stages of cuttlefish limb development (stages 16 to 19). Morphogenesis of the limb is first observed as a slight swelling (**M**) that transforms into a limb bud (**O**) as proximodistal outgrowth progresses. D, dorsal; V, ventral. (**P–S**), SEM at later stages of cuttlefish limb development (stages 21 to 25) showing the formation of sucker buds on the ventral surface of a developing limb. A primordial sucker band (yellow arrows) is observed along the ventral midline of a stage 21 limb bud (**P**). At later stages, the band cleaves superficially from the proximal end to form the sucker buds (pseudocolored blue in Q to S). Scale bars: 0.5 mm (**C and D**) and 100 μm (**M to S**).
DOI: https://doi.org/10.7554/eLife.43828.003

The following video and figure supplement are available for figure 1:

**Figure supplement 1.** Sucker morphogenesis.
DOI: https://doi.org/10.7554/eLife.43828.004
**Figure 1—video 1.** OPT 3D reconstruction showing cuttlefish hatchling morphology.
DOI: https://doi.org/10.7554/eLife.43828.005
**Figure 1—video 2.** OPT 3D reconstruction showing the internal location of the tentacles in a cuttlefish hatchling.
DOI: https://doi.org/10.7554/eLife.43828.006
**Figure 1—video 3.** OPT 3D reconstruction showing morphology of a cuttlefish embryo at stage 17.
DOI: https://doi.org/10.7554/eLife.43828.007
**Figure 1—video 4.** OPT 3D reconstruction showing morphology of a cuttlefish embryo at late stage 19.
DOI: https://doi.org/10.7554/eLife.43828.008
**Figure 1—video 5.** OPT 3D reconstruction showing the morphology of a cuttlefish embryo at stage 24.
DOI: https://doi.org/10.7554/eLife.43828.009

*Figure 1—video 4*), and by stage 24, the differential length and morphology of arms relative to tentacles is apparent (*Figure 1L*; *Figure 1—video 5*).

Analysis of sucker development showed that a sucker field primordium initially forms as a narrow proximodistal ridge along the ventral surface of each limb (evident by stage 21; *Figure 1P*). At later stages, the sucker field ridge cleaves superficially, segregating sucker buds from proximal to distal (*Figure 1Q*). As the arms elongate, the sucker buds are laid down on the entire ventral surface of each arm (*Figure 1L and R*; *Figure 1—figure supplement 1A and C–G*), forming four parallel rows across the anteroposterior axis (*Figure 1C*; *Figure 1—figure supplement 1A*). In the tentacles, the primordial sucker band is restricted to the distal tip, where sucker buds form in eight rows along the anteroposterior axis of the tentacle sucker pads (*Figure 1D*; *Figure 1—figure supplement 1B*). The full complement of immature sucker bud rows is present on each limb at hatching, and differentiation of the suckers continues during post-hatch development (*Figure 1—figure supplement 1H and I*).

## Molecular analysis of cuttlefish limbs reveals conservation of proximodistal, anteroposterior, and dorsoventral patterning networks

To test the hypothesis that cuttlefish limb development is regulated by the same molecular mechanisms that pattern arthropod and vertebrate limbs, despite their independent evolutionary origins, we cloned and characterized cuttlefish orthologs of genes that pattern the three axes of vertebrate and arthropod limbs, and then analyzed their expression patterns during cuttlefish limb development (*Figure 2* and *Figure 2—figure supplements 1–10*).

Partial sequences of cuttlefish cDNAs (*Sepia officinalis* and *Sepia bandensis*) were isolated by rt-PCR, and preliminary identities were determined by comparison with NCBI sequence databases, including the octopus genome. Molecular phylogenetic reconstructions were then made by maximum likelihood phylogenetic inference using the best amino acid substitution model for each gene

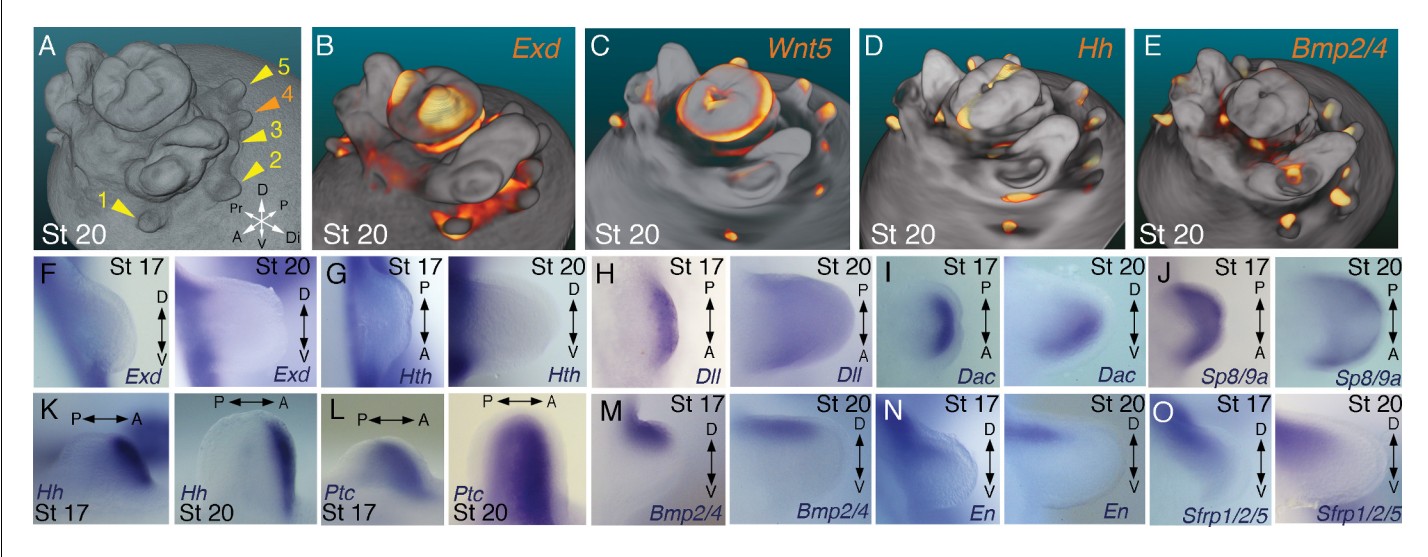

**Figure 2.** Molecular regionalization of proximodistal, anteroposterior, and dorsoventral axes during cephalopod limb development. (A) OPT reconstruction of cuttlefish embryo at stage 20 showing all five limb buds on the left side of the embryo (arms, yellow arrowheads; tentacle, orange arrowhead). (B to E) OPT reconstructions showing four representative genes with polarized expression patterns along major axes of limb buds (gene expression indicated by orange/yellow). Proximodistally polarized expression of *Exd* (B) and *Wnt5* (C). Anteroposteriorly polarized expression of *Hh* (D), dorsoventrally polarized expression of *Bmp2/4* (E). (F to O), In situ hybridizations of cuttlefish limb buds at stage 17 (left) and stage 20 (right) showing polarized patterns of expression along the proximodistal axis for *Exd* (F), *Hth* (G), *Dll* (H), *Dac* (I) and *Sp8/9a* (J); the anteroposterior axis for *Hh* (K) and *Ptc* (L); and the dorsoventral axis for *Bmp2/4* (M), *En* (N) and *Sfrp1/2/5* (O). A, anterior; P, posterior; D, dorsal; V, ventral; Di, distal; Pr, proximal.
DOI: https://doi.org/10.7554/eLife.43828.010

The following source data and figure supplements are available for figure 2:

**Source data 1.** Source data for molecular phylogenetic analyses.
DOI: https://doi.org/10.7554/eLife.43828.021
**Figure supplement 1.** Wnt phylogeny.
DOI: https://doi.org/10.7554/eLife.43828.011
**Figure supplement 2.** Pan/Tcf phylogeny.
DOI: https://doi.org/10.7554/eLife.43828.012
**Figure supplement 3.** Dac/Dach phylogeny.
DOI: https://doi.org/10.7554/eLife.43828.013
**Figure supplement 4.** Sp phylogeny.
DOI: https://doi.org/10.7554/eLife.43828.014
**Figure supplement 5.** Homeodomain transcription factor phylogeny.
DOI: https://doi.org/10.7554/eLife.43828.015
**Figure supplement 6.** Notum phylogeny.
DOI: https://doi.org/10.7554/eLife.43828.016
**Figure supplement 7.** Frizzled/Sfrp phylogeny.
DOI: https://doi.org/10.7554/eLife.43828.017
**Figure supplement 8.** Tgfβ phylogeny.
DOI: https://doi.org/10.7554/eLife.43828.018
**Figure supplement 9.** Hedgehog phylogeny.
DOI: https://doi.org/10.7554/eLife.43828.019
**Figure supplement 10.** Patched phylogeny.
DOI: https://doi.org/10.7554/eLife.43828.020

family (see Materials and methods for details). Tree topologies with well-supported bootstrap values showed the position of each cuttlefish gene within the targeted gene families, which included *Wnt*, *Tcf/Lef*, Frizzled (*Fzd*), Dachsund (*Dac/Dach*), *Notum*, Patched (*Ptc/Ptch*), Hedgehog (*Hh*), Bone morphogenetic protein (*Bmp*), Specificity protein (*Sp*), and the ANTP and TALE homeobox gene families (trees are shown in *Figure 2—figure supplements 1–10* and are described below; gene accession

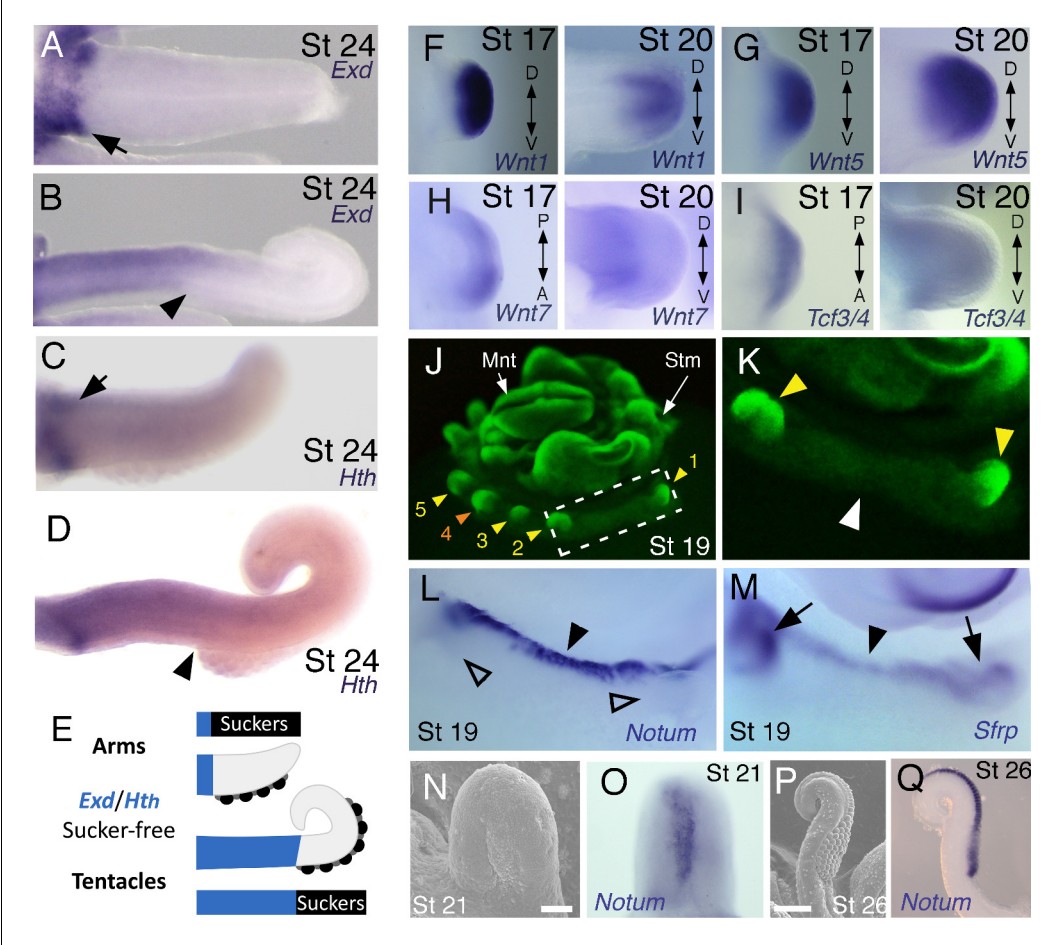

**Figure 3.** Expression of proximal identity genes *Exd* and *Hth* in arms and tentacles corresponds with distribution of suckers; Wnt signaling repressors are dorsally restricted. (**A and B**) Compared to arms (**A**), tentacles (**B**) show a distally expanded domain of *Exd* expression in the proximal region of the limb. (**C and D**) A similar pattern of expression is detected for *Hth* during arm (**C**) and tentacle (**D**) development. Distal boundary of *Exd* and *Hth* expression marked by black arrowheads in (A to D). (**E**) Expanded expression of proximal identity genes correlates with the expanded sucker-free domain seen in tentacles compared to arms. (**F and H**) The Wnt ligands *Wnt1*, *Wnt5* and *Wnt7* show a distally restricted expression but no dorsoventral polarization at stages 17 and 20. (**I**) The Wnt signaling transcription factor *Tcf3/4* is also distally restricted but shows no dorsoventral polarization at stages 17 and 20. (**J and K**) Fluorescent nuclear stain SYBR Safe highlights limb buds (yellow arrowheads). Boxed region in (**J**) is enlarged in (**K**); white arrowhead marks interlimb region. (**L and M**) The Wnt ligand repressors *Notum* and *Sfrp1/2/5* are expressed in the dorsal interlimb region (black arrowhead in L and M; compare with K). *Sfrp1/2/5* expression expands into the dorsal limb bud (black arrows in M) in stage 19 embryos, whereas *Notum* stays dorsal but proximally restricted (open arrowheads mark the limb buds in L). (**N and O**) The earliest sign of sucker formation can be detected by SEM as a slight swelling (**N**) and by *Notum* expression (**O**) on the ventral side of stage 21 limb buds. (**P and Q**) Expression of *Notum* is maintained through later stages of sucker morphogenesis, as seen in stage 26 tentacles (lateral views).

DOI: https://doi.org/10.7554/eLife.43828.022

The following source data and figure supplements are available for figure 3:

**Source data 1.** Sequence similarities of *Sepia bandensis* clones.
DOI: https://doi.org/10.7554/eLife.43828.025

**Figure supplement 1.** Expression of developmental control genes in cuttlefish limb buds.
DOI: https://doi.org/10.7554/eLife.43828.023

**Figure supplement 2.** Analysis of gene expression in the arms and tentacles of *Sepia bandensis* embryos.
DOI: https://doi.org/10.7554/eLife.43828.024

ID numbers and the data set used in the phylogenetic analyses is provided in *Figure 2—source data 1*).

Within the Wnt family of cell signaling proteins, we isolated cuttlefish orthologs of *Wnt1*, *Wnt2*, *Wnt5*, and *Wnt7* (*Figure 2—figure supplement 1*). Phylogenetic analysis of cuttlefish transcription

factors identified *Tcf3/4*, an ortholog of arthropod *Pangolin* and a pro-ortholog of vertebrate *Tcf3* and *Tcf4* (*Figure 2—figure supplement 2*), *Dac*, a pro-ortholog of vertebrate *Dach1* and *Dach2* (*Figure 2—figure supplement 3*), and *Sp8/9*, a pro-ortholog of vertebrate *Sp8* and *Sp9* (*Figure 2—figure supplement 4*). We also identified numerous homeobox genes, which phylogenetic analyses confirmed to be *Dll*, a pro-ortholog of vertebrate *Dlx* genes, *Exd*, a pro-ortholog of vertebrate *Pbx* genes, *Hth*, a pro-ortholog of vertebrate *Meis1* and *Meis2*, and *Engrailed*, a pro-ortholog of vertebrate *En1* and *En2* (*Figure 2—figure supplement 5*).

In addition, we cloned the Wnt extracellular inhibitors *Notum* and *Sfrp-1/2/5*, and the Wnt co-receptor *Fzd9/10* (*Figure 2—figure supplements 6* and *7*). Cuttlefish possess a *Bmp-2/4* gene that is an ortholog of arthropod *Dpp* and a pro-ortholog of vertebrate *Bmp2* and *Bmp4* (*Figure 2—figure supplement 8*), a *Hh* gene (*Grimaldi et al., 2008*) that we show to be a pro-ortholog of the vertebrate hedgehog family (*Figure 2—figure supplement 9*), and a gene encoding the Hh receptor *Patched*, a pro-ortholog of vertebrate *Ptch1* and *Ptch2* (*Figure 2—figure supplement 10*). The cuttlefish *Sfrp* ortholog that we identified as *Sfrp1/2/5* was annotated incorrectly in the octopus genome as *Frizzled1* (*Figure 2—source data 1*). We also found two *Sp8/9* genes in the octopus genome (*Figure 2—source data 1*), and the cuttlefish *Sp8/9* gene shows clear orthology to only one of the two octopus genes (*Figure 2—figure supplement 4*), suggesting that the *Sp8/9* gene underwent a duplication in cephalopod mollusks. Therefore, we designate the octopus *Sp8/9* paralogs as *Sp8/9a* and *Sp8/9b*, and the cuttlefish *Sp8/9* gene that we isolated is the ortholog of *Sp8/9a*.

We next investigated the spatial and temporal expression patterns of these genes during cuttlefish limb development. Genes that pattern the proximodistal axis of arthropod and vertebrate limbs (*Lecuit and Cohen, 1997*; *Mercader et al., 1999*; *Panganiban et al., 1997*; *Pueyo and Couso, 2005*) showed similarly polarized patterns of expression along the proximodistal axis of cuttlefish limb buds, with *Exd* and *Hth* restricted proximally (*Figure 2B,F and G*; *Figure 3A–E*; *Figure 3—figure supplement 1A and B*; and *Figure 3—figure supplement 2A,I and J*) and *Dll*, *Dac*, *Sp8/9a*, *Wnt1*, *Wnt5*, and *Wnt7* restricted distally (*Figure 2C,H–J*; *Figure 3F–I*; *Figure 3—figure supplement 1C–E and L–N*; and *Figure 3—figure supplement 2B,C,G,I and J*). At stages 20–21, the distal expression boundaries of *Exd* and *Hth* and the proximal expression boundaries of *Dll* and *Sp8/9a* appear to mark the morphological boundary between the proximal sucker-free and the distal sucker-forming regions (compare right panels in *Figure 2F–H and J* with *Figure 1P*). Indeed, at stages when arms and tentacles begin to develop their distinctive morphologies – tentacles are longer and have an extensive proximal sucker-free domain – the *Exd/Hth* expression domains were found to extend further distally in tentacles (*Figure 3B,D*) compared to arms (*Figure 3A and C*). This distal expansion of the *Exd/Hth* expression domain matches the expanded sucker-free region and the distal restriction of suckers in tentacles (*Figure 3E*).

Our finding that the proximodistal axis of cuttlefish limbs shares patterns of molecular regionalization with arthropod and vertebrate limbs led us to examine whether anteroposterior and dorsoventral axis development are also conserved. Posteriorly polarized activation of Hedgehog signaling in arthropod and vertebrate limbs is essential for proper patterning of the anteroposterior axis, and ectopic activation of the Hedgehog pathway induces anterior duplication of posterior structures (*Basler and Struhl, 1994*; *Kojima et al., 1994*; *Riddle et al., 1993*). We analyzed *Hh* expression during cuttlefish limb development at stages 16 to 20 and found that *Hh* expression is also polarized to one side of cuttlefish limb buds. In cuttlefishes, however, *Hh* expression is restricted to the anterior margin of the limb bud, whereas in arthropods and vertebrates, *Hh/Shh* is expressed posteriorly (*Figure 2D and K*; and *Figure 3—figure supplement 2D*). Consistent with the anterior localization of *Hh*, we detected expression of *Patched*, which serves as a readout of Hedgehog signal transduction, in an anterior-to-posterior gradient (*Figure 2L*). Thus, anteroposteriorly restricted activation of the Hedgehog pathway is a conserved feature of cephalopod, arthropod, and vertebrate limb development, but the polarity of the signaling center is reversed in cephalopod limbs. By stage 21, the anteriorly restricted *Hh* domain has diminished and a new, central expression domain appears in the location of the brachial nerve primordia (*Figure 3—figure supplement 1F,K*).

We then examined the dorsoventral axis, which is controlled by the antagonistic actions of wg/Wnt and dpp/Bmp signaling in arthropods and vertebrates (*Brook and Cohen, 1996*; *Cygan et al., 1997*; *Diaz-Benjumea et al., 1994*; *Jiang and Struhl, 1996*; *Parr and McMahon, 1995*). In arthropods, the Wnt ligand *wg* is expressed ventrally, whereas the *Bmp2/4* ortholog *dpp* is expressed dorsally (*Basler and Struhl, 1994*; *Diaz-Benjumea et al., 1994*). Expression and function of the Wnt-

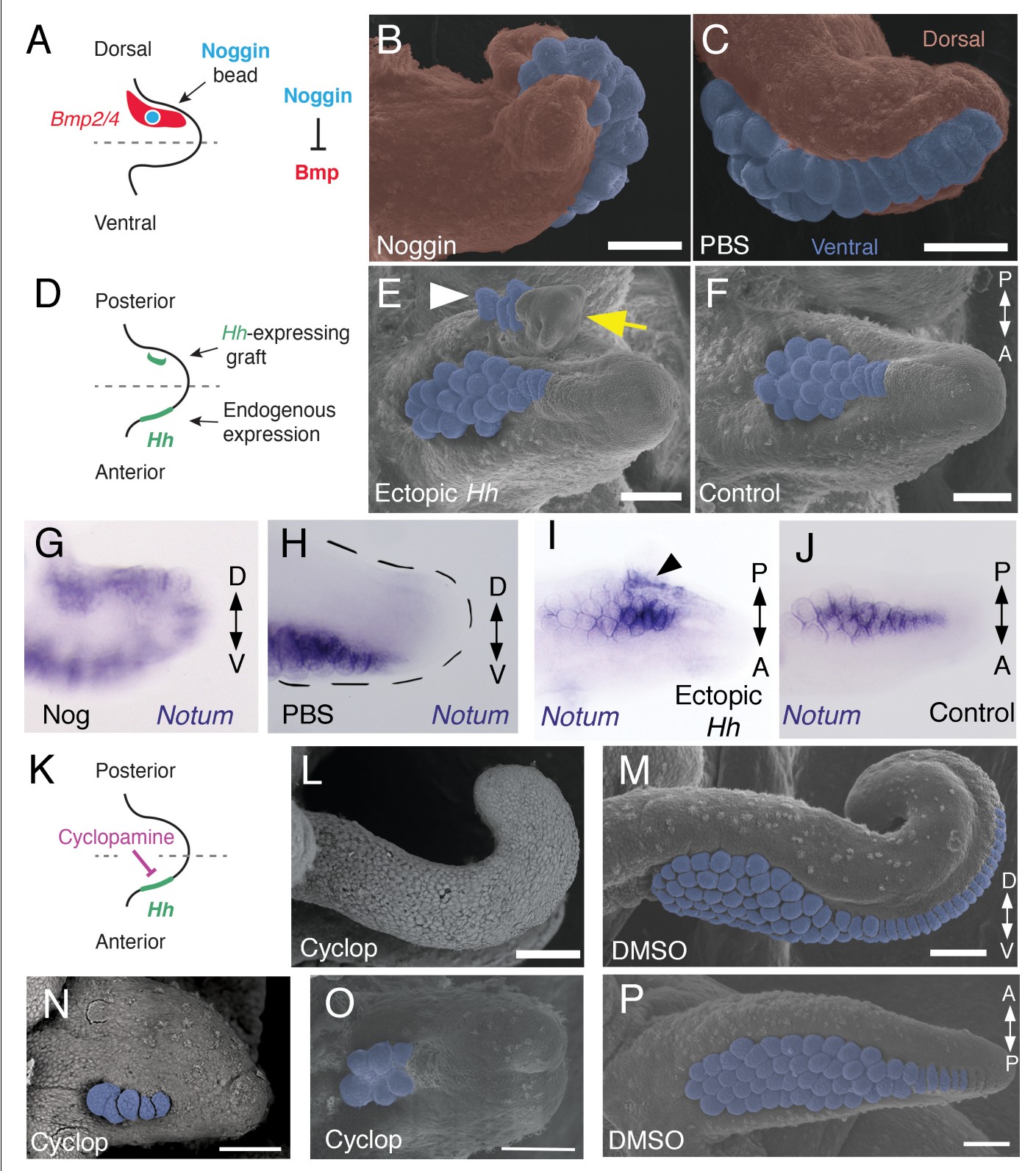

**Figure 4.** Bmp signaling controls dorsoventral patterning and Hh signaling regulates anteroposterior patterning of cuttlefish limbs. (**A** to **C**) Implantation of carrier beads loaded with the Bmp inhibitor Noggin (**A**) results in formation of ectopic sucker buds (n = 8/12) on the dorsal surface of the limb (**B**), whereas PBS control beads (n = 15/15) result in normal development of the dorsal limb (pseudocolored red) restriction of suckers (pseudocolored blue) to the vental side of the limb (**C**). (**D-F**), Hh-expressing cells (taken from the funnel of a stage 24 donor embryo) grafted to the

*Figure 4 continued on next page*

*Figure 4 continued*
posterior side of a stage 17 cuttlefish limb bud (**D**) generates a posterior mirror-image limb duplication (n = 7/12; yellow arrow in E), whereas no duplication (n = 8/8) results when control (Hh-negative) cells are grafted to the same position (**F**). Sucker buds are pseudocolored blue in (**E and F**); sucker buds in duplicated limb marked with a white arrowhead. (**G and H**) Noggin beads induce ectopic expression of *Notum* on the dorsal side of the limb (**G**). Limbs receiving control PBS beads show normal expression of *Notum* ventrally (**H**). (**I and J**) Graft of *Hh*-expressing tissue to the posterior side of the limb induces ectopic domain of *Notum* prior to duplication of the limb (**I**; black arrowhead). Note the two separate domains of *Notum* expression in I compared to a single *Notum* expression domain in the limb with the *Hh*-negative control graft (**J**). (**K to P**) Transitory repression of Hh signaling by cyclopamine (**K**) during early stages of limb development disrupts the anteroposterior distribution of sucker bud rows (**L, N, O**). Cyclopamine-treated limbs showing complete loss of suckers in tentacles (**L**) and reduction in the number of sucker bud rows in arms (**N and O**). Control embryos treated with vehicle only (DMSO) develop the normal number of sucker bud rows in tentacles (**M**) and arms (**P**). Sucker buds are pseudocolored blue in (**B, E, F, and M- P**). Scale bars 100 µm.
DOI: https://doi.org/10.7554/eLife.43828.026
The following figure supplement is available for figure 4:

**Figure supplement 1.** Sucker development after manipulations of Bmp and Hh signaling pathways.
DOI: https://doi.org/10.7554/eLife.43828.027

Bmp network is conserved, albeit with inverted polarity, in vertebrate limbs; *Wnt7a* is expressed dorsally (*Parr and McMahon, 1995*) and Bmp signaling activates *Engrailed1* (*En1*) ventrally (*Ahn et al., 2001*), and these interactions regulate development of dorsal and ventral limb structures (*Cygan et al., 1997*; *Parr and McMahon, 1995*). During cuttlefish limb development, *Bmp2/4* and *En* show dorsally polarized expression (*Figure 2E,M and N*; and *Figure 3—figure supplement 2E*). Genes encoding Wnt ligands (*Wnt1*, *Wnt5* and *Wnt7*) and cellular components of canonical Wnt signaling cascade (*Tcf3/4* and *Frz9/10*) are expressed broadly throughout the dorsoventral axis of cuttlefish limb buds (*Figure 3F–I* and *Figure 3—figure supplement 1L–R*; and *Figure 3—figure supplement 2G,I and J*); however, the secreted Wnt antagonists *Notum* and *Sfrp1/2/5* are expressed dorsally in the limb and interlimb regions (*Figure 3J–M*), with the *Sfrp1/2/5* domain extending deeper into the dorsal limb buds (*Figure 2O*; *Figure 3M*). This dorsal expression of Wnt antagonists suggests a mechanism for restriction of Wnt signaling to the ventral side of the cephalopod limb buds. Taken together, these results suggest that the genetic pathways active along the proximodistal, anteroposterior, and dorsoventral axes of cephalopod limbs are homologous (specifically, orthologous) to the networks that regulate limb development in arthropods and vertebrates.

In order to further test this hypothesis, we next performed a series of functional experiments to determine whether polarized expression of these signaling molecules is involved in patterning the anteroposterior and dorsoventral axes of cuttlefish limbs (described below). We developed a method for *ex-ovo* culture of cuttlefish embryos (see Material and methods) to allow in vivo manipulations of genetic pathways in early limb buds.

## Bmp signaling controls dorsoventral patterning of cuttlefish limbs

A hallmark of dorsoventral polarity is the restriction of sucker buds to the ventral surface of the limb (*Figure 1C,D and S*), and this is preceded by ventral expression of *Notum* in the sucker-forming region at stage 21 (*Figure 3N–Q*). We asked whether polarized expression of *Bmp2/4* on the dorsal side of cuttlefish limb buds is required for the specification of dorsal identity. To repress dorsal Bmp activity, we implanted carrier beads loaded with Noggin (Nog), a secreted Bmp inhibitor protein, on the dorsal side of stage 17 limb buds (*Figure 4A*). Implantation of Nog beads on the dorsal side of cuttlefish limb buds resulted in ectopic, dorsal expansion of the *Notum* mRNA domain (n = 3/3; control PBS [phosphate buffered saline] beads had no effect on *Notum* expression [n = 3/3]) (*Figure 4G,H*). To determine whether inhibition of dorsal Bmp signaling respecifies dorsal cells to form ventral structures, we repeated the experiment and allowed embryos to develop to stage 26–27. Analysis of limb morphology by scanning electron microscopy revealed the presence of ectopic sucker buds on the dorsal surface of Nog-treated limbs (n = 8/12; *Figure 4B*; *Figure 4—figure supplement 1A and B*). The ectopic dorsal suckers extended around the distal tip of the limb and joined the ventral sucker field. By contrast, in limbs that received control PBS beads dorsally, sucker buds were restricted to ventral surface and terminated at the normal dorsal-ventral boundary at the tip of the limb (n = 15/15; *Figure 4C*). Our finding that antagonism of Bmp signaling results in

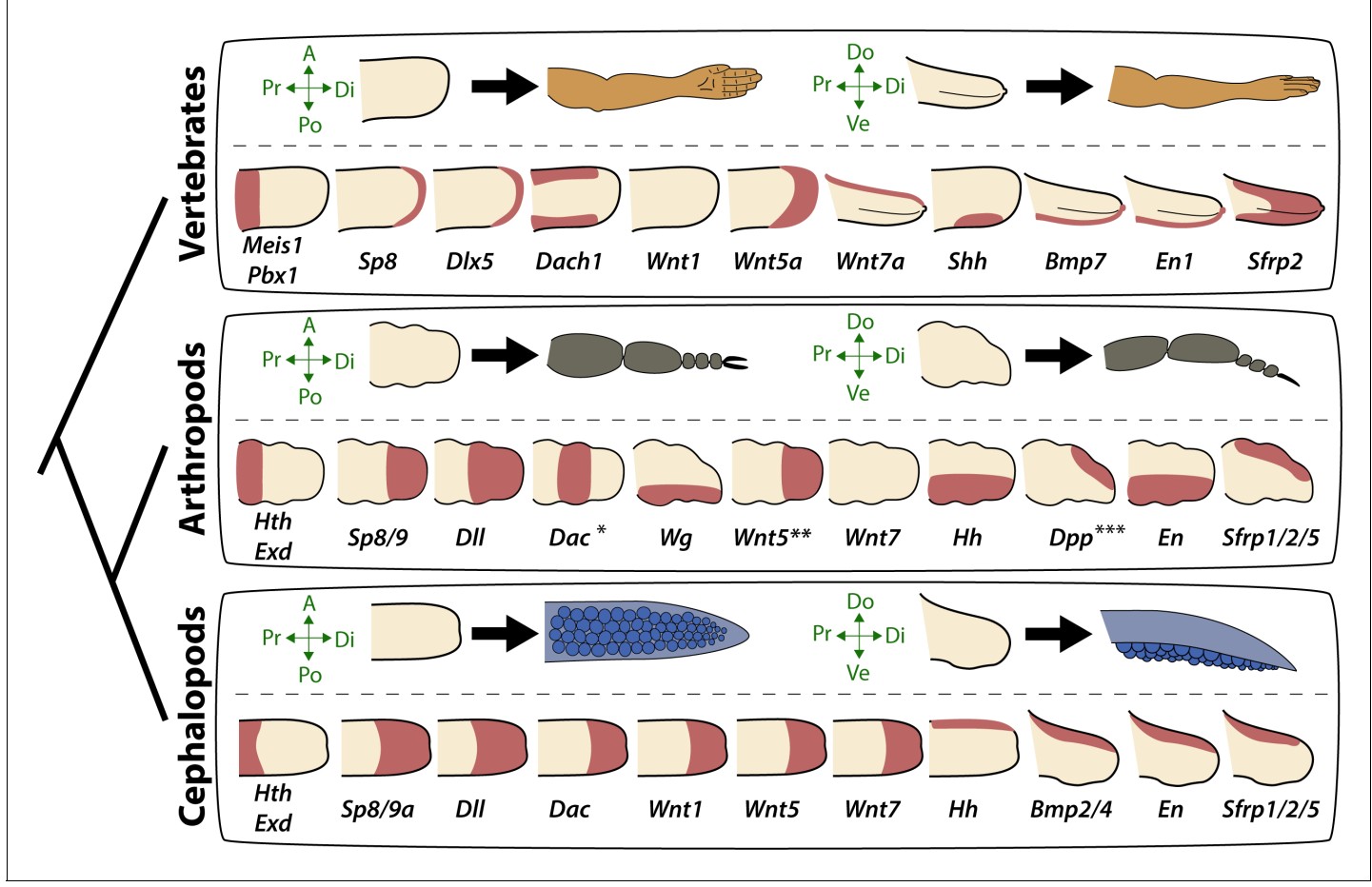

**Figure 5.** Molecular patterning of the anteroposterior, proximodistal, and dorsoventral axes of developing limbs in vertebrates, arthropods, and cephalopods. For each lineage, the top row shows schematic representations of a generalized limb bud and an adult limb in two different orientations. Axes are indicated to the left of each limb bud (A, anterior; Po, posterior; Pr, proximal; Di, distal; Do, dorsal; Ve, ventral) Bottom rows show limb buds with gene expression domains pink color). Vertebrate gene expression based on mouse limb development. Arthropod gene expression is a compound reconstruction from chelicerate, myriapod, and hexapod limb development in order to consolidate a complete set of pro-orthologous genes comparable to vertebrate and cephalopod lineages. Cephalopod gene expression is based on findings in this study from the cuttlefish *Sepia officinalis* and *Sepia bandensis*. The figure illustrates the conserved and divergent expression patterns of homologous (orthologous) genes, some of which share equivalent roles in patterning the limb axes. The proximodistal axis displays conserved expression of transcription factors at opposite ends; *Hth* (pro-ortholog of vertebrate *Meis* genes) and *Exd* (pro-ortholog of vertebrate *Pbx* genes) are restricted proximally, whereas *Dll* (pro-ortholog of vertebrate *Dlx* genes), *Wnt5* (pro-ortholog of *Wnt5a*) and *Sp8/9* (pro-ortholog of vertebrate *Sp8* and *Sp9* genes, known as *Sp6-9* in some arthropods) show distally restricted expression. The typical expression pattern of *Dac* seen in arthropods (between proximally and distally restricted genes) is not strictly conserved in vertebrates (*Dac* is the pro-ortholog of vertebrate *Dach* genes) or cephalopods. However, *Dac* expression in non-locomotory arthropod appendages (e.g., mandibles) is distally restricted, resembling cephalopod *Dac* expression (**Donoughe and Extavour, 2016**). Expression patterns of the diverse family of *Wnt* genes shows interesting variation. Although, some members of the family show variation in their expression pattern (*Wnt1* and *Wnt7*), there is a general pattern of distal restriction of Wnt expression (represented here by *Wnt5*, but also seen in many other Wnt ligands) in the three lineages. At the level of individual Wnt members, *Wg* (pro-ortholog of vertebrate *Wnt1*) is restricted ventrally in arthropods but not in vertebrates or cephalopods, and *Wnt7a* (arthropod and cephalopod *Wnt7* genes are pro-orthologs of vertebrate *Wnt7a*) is restricted dorsally in vertebrates but not in arthropods or cephalopods. Restricted expression of Wnt ligands either dorsally or ventrally has not been reported in cephalopods, but the dorsally restricted expression of the Wnt repressor *Sfrp1/2/5* suggests a role of polarized Wnt pathway activation in the control of the dorsoventral axis of cephalopod limbs, similar to vertebrates (by dorsal *Wnt7a*) and arthropods (by ventral *Wg*). There is a clear restriction of at least one Bmp ligand (vertebrate *Bmp7* and cephalopod *Bmp2/4*; pro-orthologs of arthropod *Dpp*) and the transcription factor *En* along the dorsoventral axis in these three lineages. Finally, polarized expression of *Hh* is conserved in the three lineages (posterior in vertebrates and arthropods, but anterior in cephalopods), which, together with the functional manipulations, indicates conservation of Hh signaling in patterning the anteroposterior limb axis in the three lineages. The asterisk (*) in arthropod *Dac* indicates that some mouth appendages show a distal expression domain (**Donoughe and Extavour, 2016**) (**Angelini and Kaufman, 2005**) more similar to cephalopod *Dac* limb expression than to *Dac* expression in arthropod legs. Two asterisks (**) indicate that *Wnt5* expression shows variation in arthropods, with a sub-distal expression in chelicerates (**Damen, 2002**) but distal in hexapods (i.e. flour beetle) (**Bolognesi et al., 2008**). Three asterisks (***) indicate that *Dpp* shows variation in its expression domain in arthropods, with some hexapods and

*Figure 5 continued on next page*

*Figure 5 continued*

chelicerates showing a distal expression domain, whereas in Myriapods and other hexapods it is dorsally restricted, as depicted here (*Angelini and Kaufman, 2005*). Schematized gene expression domains for vertebrates and arthropods are from the following sources. Mouse gene expression: *Meis1* (*González-Lázaro et al., 2014*), *Pbx1* (*Capellini et al., 2006*), *Sp8* (*Kawakami et al., 2004*), *Dlx5* (*Vieux-Rochas et al., 2013*), *Dach1* (*Salsi et al., 2008*), *Wnt, Wnt5a, Wnt7a, Sfrp2* (*Witte et al., 2009*), *Shh* (*Riddle et al., 1993*), *Bmp7* (*Choi et al., 2012*) and *En1* (*Loomis et al., 1998*). Arthropod expression based on: Chelicerates, *Hth, Exd, Dll, Dac* (*Prpic et al., 2003*), *Sp8/9* (*Königsmann et al., 2017*), *Wg* (*Damen, 2002*), *En* (*Damen, 2002*) and *Sfrp1/2/5* (*Hogvall et al., 2018*); Myriapods, *Hth, Exd, Dll, Dac* (*Prpic and Tautz, 2003*), *Sp8/9* (*Setton and Sharma, 2018*), *Wg, Dpp* (*Prpic, 2004*), *Wnt5, Wnt7, Hh, En* (*Janssen et al., 2004*), *Sfrp1/2/5* (*Hogvall et al., 2018*); Hexapods, flour beetle, *Hth, Exd* (*Prpic et al., 2003*), *Sp8/9* (*Schaeper et al., 2010*), *Dll* (*Beermann et al., 2001*), *Dac* (*Prpic et al., 2001*), *Wg, Wnt5* (*Bolognesi et al., 2008*), *Dpp* (*Sanchez-Salazar et al., 1996*), *En* (*Brown et al., 1994*); Hexapods, cricket, *Hth, Exd, Dll, Dac, Wg, Hh, Dpp, En* (*Donoughe and Extavour, 2016*).
DOI: https://doi.org/10.7554/eLife.43828.028

The following figure supplement is available for figure 5:

**Figure supplement 1.** The developing funnel/siphon organ shows limb-like expression patterns of the proximodistal patterning genes *Exd* and *Wnt5*.
DOI: https://doi.org/10.7554/eLife.43828.029

development of ventral structures (sucker buds) on the dorsal side of the limb indicates that dorsal *Bmp2/4* activity is required for the early specification of dorsal identity in cephalopod limb development.

## Hedgehog signaling at the anterior margin of cuttlefish limb buds controls anteroposterior patterning of the sucker field

We then investigated whether the mechanism of anteroposterior patterning is conserved between cephalopod and vertebrate/arthropod limbs. To determine whether the anterior expression of *Hh* in cuttlefish limb buds controls anteroposterior patterning, we grafted *Hh*-expressing cells from the thickened funnel epithelium (*Tarazona et al., 2016*) to the posterior side of stage 17 limb buds, which created an ectopic source of Hh opposite the endogenous *Hh* expression domain (*Figure 4D*). We used *Hh*-expressing cells from the funnel, rather than the anterior side of the limb bud, to exclude the possibility of grafted limb cells undergoing self-differentiation. Transplantation of *Hh*-expressing cells to the posterior side of cuttlefish limb buds resulted in posterior limb duplications (n = 7/12; *Figure 4E* and *Figure 4—figure supplement 1C,D*). Analysis of morphology and gene expression in host limbs approximately 10 days after receiving the graft revealed that the posterior duplications even contained sucker buds, which were marked by *Notum* expression (*Figure 4I and J*). By contrast, limbs that received control grafts of stage 24 funnel epithelium that lacks *Hh* expression (*Tarazona et al., 2016*) developed normally (n = 8/8; *Figure 4F*).

Although these results suggest that *Hh* is sufficient to re-specify anteroposterior polarity in cuttlefish limbs, we wanted to exclude the possibility that posterior identity was induced by other factors that could be present in the graft. Therefore, we tested whether Hh signaling is necessary for anteroposterior patterning of cephalopod limbs by specifically repressing endogenous Hh signaling. A notable morphological feature of cephalopod limbs is the anteroposterior arrangement of parallel sucker rows on the ventral surface (*Figure 1C,D and S*). Based on the results of the transplantation experiments, we reasoned that Hh signaling could regulate the number of sucker rows along the anteroposterior axis of cephalopod limbs, similar to the manner in which Hh specifies digit number along the anteroposterior axis of vertebrate limbs (*Lewis et al., 2001*; *Scherz et al., 2007*; *Zhu et al., 2008*).

Transitory treatment (2 days) of cuttlefish embryos at stage 16, when *Hh* is first expressed on the anterior side of the early limb bud, with the small molecule cyclopamine, an inhibitor of Smoothened that represses Hh signaling (*Figure 4K*), disrupted the anteroposterior distribution of sucker rows in arms and tentacles. Severity of this phenotype ranged from arms with a reduced number of suckers and sucker rows (n = 10/10; *Figure 4N and O*) to completely sucker-free tentacles (n = 8/10; *Figure 4L*). Control treatments with vehicle only (DMSO) did not alter the normal anteroposterior pattern of sucker rows (n = 8/8; *Figure 4M and P*). Finally, to confirm that the phenotype of cyclopamine-treated embryos was not due to failure in brachial nerve differentiation, we examined acetylated tubulin immunofluorescence, which shows that the brachial nerve cords develop in both cyclopamine and DMSO treated embryos (*Figure 4—figure supplement 1E,F*). These results show

that Hh signaling is necessary for proper patterning of the anteroposterior axis in cephalopod limb development.

## Discussion

Our finding that the proximodistal, dorsoventral, and anteroposterior axes of cuttlefish limb buds are patterned by the same pathways that regulate arthropod and vertebrate limb development suggests that the independent evolution of limbs in cephalopod mollusks involved recruitment of an ancient genetic program for appendage development. Discovery of this appendage developmental circuit within Spiralia demonstrates its deep conservation across all three branches of Bilateria (i.e., Deuterostomia, Ecdysozoa, and Spiralia), suggesting its presence in the common ancestor of all bilaterians (*Figure 5*). Parallel recruitment of this ancient developmental genetic program may have played a role in the independent evolution of a wide diversity of appendages in many bilaterian lineages (*Moczek and Nagy, 2005*; *Shubin et al., 2009*).

The discovery that cephalopod, arthropod, and vertebrate appendages develop using conserved developmental mechanisms does not exclude the possibility that other types of appendages evolved by recruiting a different set of developmental tools (or by utilizing the same tools but in different patterns). Examination of gene expression in lateral parapodial appendages of the polychaete worm *Neanthes,* also a spiralian, led to the suggestion that the molecular mechanisms of polychaete appendage development might not be conserved with ecdysozoans and deuterostomes (*Winchell and Jacobs, 2013*; *Winchell et al., 2010*). However, given that relatively few genes were examined in *Neanthes* parapodia, it is difficult to conclude whether the reported differences between parapodia and arthropod/vertebrate/cephalopod limbs reflect the unique nature of parapodia or lineage-specific divergences that occurred after recruitment of the core developmental program. A study of a different polychaete, *Platynereis dumerilii,* showed that gene expression is generally conserved in appendages that form during regeneration of caudal trunk segments, although some divergent patterns were observed and these were suggested to reflect taxon-specific differences in appendage morphology (*Grimmel et al., 2016*). How parapodia fit into the picture of animal appendage evolution will require additional studies of spiralian appendages to increase the diversity of species, types of appendages, and number of genes/pathways interrogated. Nonetheless, our discovery that cephalopod arms and tentacles evolved by parallel recruitment of the same genetic program that orchestrates appendage formation in arthropods and vertebrates suggests that this program was present in the bilaterian common ancestor.

Activation of this ancient developmental program could also underlie the origin of other morphological innovations, including non-locomotory appendages such as beetle horns (*Moczek and Nagy, 2005*; *Moczek et al., 2006*) and external genital organs of amniote vertebrates (*Cohn, 2011*; *Gredler et al., 2014*). We propose that the genetic program for appendage formation was stabilized in Bilateria, including those lineages that lack limbs, for development of appendage-like structures. This hypothesis implies that the ancestral appendage developmental program was not a latent developmental feature that was redeployed each time that limbs evolved, but rather it might have been a continuously activated network that controlled formation of outgrowths in general.

One of our observations raises the possibility that the gene network that controls appendage formation could be conserved in non-cephalopod mollusks, despite the absence of arms and tentacles in those lineages. During cuttlefish funnel/siphon development, we found asymmetric expression of *Hh* (*Tarazona et al., 2016*) and proximodistally polarized expression of *Wnt5* and *Exd*, which partially mirror their expression patterns during arm and tentacle development (*Figure 5—figure supplement 1*). If this gene network is found to be active in the developing funnel/siphon of non-cephalopod mollusks, then the funnel/siphon would represent a more primitive site of expression in mollusks, given that evolution of the molluscan funnel/siphon predates the origin of cephalopod limbs (*Nielsen, 2012*; *Ruppert et al., 2004*). Further studies of gene expression and function during funnel/siphon development in mollusks will be needed to determine if this clade shows conservation of the appendage development program beyond cephalopod arm and tentacle development.

Although the bilaterian common ancestor may have used this genetic program to control development of rudimentary outgrowths (e.g., appendages, funnel/siphon, genitalia), it is also possible that it predates the evolution of locomotory and non-locomotory appendages. Studies of cephalic neuroectoderm showed that gene expression patterns controlling the anteroposterior axis of the

neuroectoderm mirror the organization of gene expression territories along the proximodistal axis of locomotory appendages, including polarized expression of *Sp8*, *Dll*, *Dac* and *Hth* (**Lemons et al., 2010**). Similarly, Minelli has suggested that the appendage patterning program could reflect co-option of a more ancient (pre-bilaterian) program for patterning the main body axis and, therefore, bilaterian appendages are simply secondary body axes (**Minelli, 2000**; **Minelli, 2003**).

Cephalopod arms and tentacles have no direct structural homologs in non-cephalopod mollusks; however, they likely formed from the ventral embryonic foot, a morphological and embryological hallmark of the molluscan bodyplan (**Nödl et al., 2016**). Therefore, cephalopod arms and tentacles may be considered evolutionary novelties that are derived from a structure that is conserved across Mollusca. This raises the question of whether other foot-derived outgrowths/appendages (e.g., in sea slugs) evolved by co-option of the same developmental program that cephalopods, arthropods, and vertebrates use to build appendages.

Although the results presented here suggest that an ancient and conserved developmental genetic program facilitated the origin of cephalopod limbs, they also indicate that fine-scale regulatory changes may have played a role in the diversification of cephalopod limb morphologies. For example, evolution of specialized tentacles from serially homologous arms may have resulted from a distal shift in the expression of proximal identity genes, such as *Exd* and *Hth,* which could have extended the proximal sucker-free domain and restricted suckers to a distal pad (*see Figure 3A–E*). Likewise, the results of functional manipulations of Hh signaling in cuttlefish limbs suggests that the diversity in the number of sucker rows in cephalopod limbs (i.e. four rows in squids and cuttlefishes, two in octopus, and one in vampire squid and glass octopus) could be explained by modulation of Hh signaling, in the same way that gradual changes to *Shh* regulation has led to variation in digit number in tetrapod vertebrates (**Scherz et al., 2007**; **Shapiro et al., 2003**; **Zhu et al., 2008**).

Finally, we note that while the data presented here point to the existence of a deeply conserved genetic program for appendage development across *Bilateria*, this does not imply that the limbs of cephalopods, arthropods, and vertebrates are homologous structures, or that limbs were present in the common ancestor. Rather, these results show that homologous developmental mechanisms underlie the multiple parallel origins of limbs in bilaterians.

## Materials and methods

No statistical methods were used to predetermine sample size. Embryos were randomized in each experiment. The investigators were not blinded to allocation during experiments and outcome assessment.

### Embryo collection and preparation

*Sepia officinalis* and *Sepia bandensis* eggs were purchased from commercial suppliers, incubated until they reached the required stages (**Lemaire, 1970**), and prepared for in situ hybridization (ISH) and immunohistochemistry as described (**Tarazona et al., 2016**).

### Optical projection tomography (OPT)

Three-dimensional reconstructions of gene expression in cuttlefish embryos were performed as previously described (**Tarazona et al., 2016**).

### Scanning electron microscopy

Cuttlefish embryos were fixed in 4% paraformaldehyde in phosphate buffered saline (PBS) overnight at 4°C and were washed with PBS the next day. Embryos were fixed in 1% osmium tetroxide solution in PBS for 30 min and then washed three times in PBS, dehydrated through a graded ethanol series, critical point dried, and sputter coated with gold. Embryonic samples were scanned using a Hitachi SU5000 and Hitachi TM3000.

### Gene cloning and molecular phylogenetic analysis

RNA extraction from *Sepia officinalis* and *Sepia bandensis* embryos at stages 15–26 was performed using TRIzol reagent (Ambion) following the manufacturer's instructions. cDNA synthesis was performed by an AMV reverse transcriptase (New England Biolabs) following the manufacturer's

instructions. PCR amplification was carried out on *Sepia* cDNA pools, amplicons were cloned into TA vectors and sequenced. We then performed multiple sequence alignments (MSA) with ClustalW (PMID: 7984417) using the predicted amino acid sequence of our cuttlefish cDNA fragments, and putative metazoan orthologous genes downloaded from NCBI RefSeq protein databases (*Figure 2—source data 1*). We performed nine MSA for Wnt, Tcf, Sfrp, Notum, Patch, Hh, Bmp, Sp and Homeo-domain families. Each of the nine MSA was analyzed by ProtTest (PMID: 15647292), in order to determine the best combination of amino acid substitution model and other free parameters (amino acid site frequency, site heterogeneity and invariant sites), using Akaike information criterion (*Figure 2—source data 1*). We applied the best model in RaXML (PMID: 18853362) for each MSA and performed maximum likelihood phylogenetic inference, estimating branch support by bootstrap, and then majority consensus of the trees from all bootstrap partitions was performed to compute the final tree topology. All sequences have been deposited in Genbank under accession numbers MK756067-MK756082 (complete list of entries is provided in *Figure 2—source data 1*).

## In situ hybridization (ISH) and immunohistochemistry

Whole-mount ISH was performed using digoxigenin- and fluorescein-labeled antisense (or sense control) RNA probes according to protocols described previously (*Tarazona et al., 2016*). Due to limited availability of embryonic material at relevant early developmental stages, only a limited number of *S. bandensis* embryos were used for ISH. Thus, the majority of ISH were performed in *S. officinalis* embryos using *S. officinalis* antisense RNA probes, however, some ISH were performed in *S. officinalis* embryos using *S. bandensis* antisense RNA probes. We validated the specificity of *S. bandensis* probes in *S. officinalis* embryos by comparing the gene expression domains marked by these probes in embryonic material from both species at stages 20 and 21. This comparison shows that gene expression territories identified by these probes at these stages were indistinguishable between the two species (*Figure 3—figure supplement 2*), consistent with their high level of sequence similarity (*Figure 3—source data 1*). Excluding the *S. bandensis* ISH mentioned above, all the experiments described in this work were carried out with *S. officinalis* embryos. Proliferating cells were detected by immunolocalization of Histone H3 Serine 10 phosphorylation using an antibody against H3S10p/PHH3 (06–570, EMD Millipore) and brachial nerve tissue was detected using an antibody against acetylated alpha tubulin (ab24610, Abcam).

## Cuttlefish *ex-ovo* embryo culture and embryo manipulations

A protocol for *ex-ovo* cuttlefish embryo culture was established for this study, as a modified version of previous descriptions of *ex-ovo* embryo culture in squid (*Arnold, 1990*). Briefly, to minimize the problem of bacterial and fungal contamination we started the protocol by taking 10 cuttlefish eggs at the appropriate stage, placing them in a 50 ml tube, and washing them with 0.22 μm filtered artificial sea water (FASW) five times. Eggs were then cleaned with a freshly prepared 5% bleach solution (0.25% sodium hypochlorite in FASW) for 5 s and immediately washed with FASW five times. The bleaching and washing steps were repeated two to three times. Five additional washes with FASW were carried out before incubating the eggs in 2X antibiotic/antimycotic solution (A5955, Sigma) in FASW for 2 hr at ambient temperature.

Each cuttlefish egg was then transferred to a 50 mm diameter petri dish that was coated with a ~ 5 mm layer of 0.5% low melting point agarose (16520050, ThermoFisher), and filled with culture medium (components described below). The agarose layer had a hemispherical depression in the center of the dish made with a sterile 10 mm acrylic necklace bead before gel solidification. The 10 mm hemispherical depression is essential to maintain the normal shape of the yolk mass once the embryos are outside their egg case. Embryos were then extracted from their egg cases (*S. officinalis* are housed individually, one embryo per egg case) very slowly and with extreme care to avoid rupturing the yolk mass at the vegetal pole of the egg and were carefully placed in the hemispherical depression in the agarose. To extract the embryo, a single 5 mm diameter hole was created in the egg case, which generates a burst of the vitelline liquid and part of the embryo out from the egg case. With the hole kept open, the spontaneous shrinkage of the egg case aided in the expelling of the large cuttlefish embryo. Of every ten eggs prepared this way, between two and five embryos were damaged and had to be discarded. Embryos were cultured at 17°C.

## Protein carrier beads and tissue grafting

For protein carrier bead implantation, 150 µm diameter Affi-Gel Blue Gel beads (153–7301, Biorad) were selected and transferred to 1 mg/ml recombinant human Noggin protein (6057 NG, R and D Systems) in PBS and incubated for 30 min to 1 hr at ambient temperature before being implanted in embryos. Control beads were incubated in PBS only.

Grafts of *Hh*-expressing tissue were performed using stage 24 donor embryos and carefully dissecting the funnel side of the mantle-funnel locking system, which carries the *Hh*-expressing thickened funnel epithelium (*Tarazona et al., 2016*). The dissected tissue was transferred to 10 mg/ml Dispase II (D4693, Sigma) in cuttlefish culture medium and incubated for 40 min or until the thickened epithelium was easily detaching from the underlying mesenchyme with the aid of forceps. Tissue was then transferred to cuttlefish culture medium without Dispase II, where they were washed and then grafted into limb buds of stage 17 host embryos. Control grafts were performed using the non-*Hh* expressing epithelium of the funnel.

After bead implantation or tissue grafts, embryos were incubated at 17°C until control embryos reached stage 26, at which point all embryos were collected and prepared for SEM or ISH.

## Cuttlefish culture medium

We used a modified version of a cell culture medium for squid neuron, glia and muscle cells that was previously described (*Rice et al., 1990*). Cuttlefish culture medium had no glucose, was buffered with 20 mM HEPES and adjusted the pH to 7.6. The medium contained: 430 mM NaCl, 10 mM KCl, 10 mM CaCl2, 50 mM MgCl2, 1X MEM Non-Essential Amino Acids Solution (11140–076, Life Technologies), 1X MEM Amino Acids Solution (11130–051, Life Technologies), 1X MEM Vitamin Solution (11120–052, Life Technologies), 2 mM L-Glutamine (25030–081, Life Technologies). The medium was supplemented with 20% heat inactivated fetal bovine serum (16000044, ThermoFisher) and 1X antibiotic/antimycotic solution (A5955, Sigma).

## Treatments with small-molecule inhibitors

Cyclopamine treatments were performed as described previously (*Tarazona et al., 2016*) with the following modifications; stage 16 embryos were treated with 10 µM cyclopamine (C988400, Toronto Research Chemicals) for 2 days, then washed thoroughly ten times with FASW. Embryos were then washed five more times every hour and one time every day before collecting the embryos for SEM. Control embryos were treated with 0.1% DMSO and then washed as described above.

## Acknowledgements

We thank Emily Merton for technical support, Karen L Kelley and Kimberly L Backer-Kelley (UF ICBR) for assistance with electron microscopy, and members of our laboratory for helpful comments and discussions. OAT was supported by a Howard Hughes Medical Institute International Student Research Fellowship, DHL by a Society for Developmental Biology 'Choose Development!' fellowship, and LAS by an EDEN Undergraduate Internship.

## Additional information

### Funding

| Funder | Grant reference number | Author |
| --- | --- | --- |
| Howard Hughes Medical Institute | | Oscar A Tarazona Davys H Lopez Leslie A Slota Martin J Cohn |
| Howard Hughes Medical Institute | International Student Research Fellowship | Oscar A Tarazona |
| Society for Developmental Biology | 'Choose Development!' fellowship | Davys H Lopez |
| National Science Foundation | EDEN Undergraduate Internship | Leslie A Slota |

The funders had no role in study design, data collection and interpretation, or the decision to submit the work for publication.

## Author contributions
Oscar A Tarazona, Conceptualization, Formal analysis, Validation, Investigation, Visualization, Methodology, Writing—original draft; Davys H Lopez, Leslie A Slota, Validation, Investigation; Martin J Cohn, Conceptualization, Resources, Supervision, Funding acquisition, Validation, Project administration, Writing—review and editing

## Author ORCIDs
Leslie A Slota (iD) https://orcid.org/0000-0001-6911-811X
Martin J Cohn (iD) https://orcid.org/0000-0002-5211-200X

## Decision letter and Author response
Decision letter https://doi.org/10.7554/eLife.43828.031
Author response https://doi.org/10.7554/eLife.43828.032

# Additional files

## Data availability
All data generated or analyzed during this study are included in the manuscript and supporting files. Sequence data have been deposited in GenBank under accession ID numbers MK756067-MK756082 and are provided, together with source data for multiple sequence alignments, in Figure 2—source data 1.

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
