## [Decision Letter]

Thank you for submitting your work entitled "Evolution of limb development in cephalopod mollusks" for consideration at *eLife*. Your article has been favorably evaluated by Diethard Tautz (Senior Editor), a Reviewing Editor, and two reviewers.

The reviewers have discussed the reviews with one another and the Reviewing Editor has drafted this decision to help you prepare a revised submission.

1) Please consider extending the Discussion along the following guidelines:

a) Consider discussing a possible ancestral function of the conserved patterning module identified in your work, if the original function was not limb patterning (suggested by reviewer #1). Parts of the gene patterning functions are actually found in phylogenetically older animal groups: could these mirror aspects of the original function of the patterning module? (suggested by reviewer #2).

b) Consider including a schematical summary of the gene expression patterns in the cephalopod compared with vertebrates and arthropods, to make it easier for readers to appreciate the similarities between these animal groups (suggested by reviewer #2).

2) In most of the phylogenetic trees shown in the supplementary data there are the sequences for *S. bandensis* and not *S. officinalis*. Please clarify why there are sequences from two different species, when the cloning and further analysis was only carried out in *S. officinalis*. (suggested by reviewer #2).

3) Please also revise the following points (combined from the suggestions by both reviewers):

a) You mention in your manuscript that there are no other mollusks that have appendages. Could the captacula of Scaphopods be homologous to cephalopod tentacles? If so, this could be worth mentioning.

b) Discussion: expression patterns of *hth, exd, Dll*, and *dpp* can be found for *Platynereis* during caudal regeneration in Grimmel et al., 2016 (doi: 10.1186/s13227-016-0046-6).

c) Figure 1G: the legend for the DNA counterstaining should be called Hoechst.

d) Figure 2: each panel of the composite should have its own label (same for Figure 3F-I). For better overview, all limb buds should be shown in the same orientation. In some in situ pictures it is quite hard to see the general shape of the limb buds. Here, a dotted outline would make the pictures much clearer (similar to Figure 4H) (same for Figure 3 F-I).

Please find below the full comments by both reviewers, for your information:

*Reviewer #1:*

I congratulate the authors to an outstanding contribution to evolutionary developmental biology. The authors demonstrate that cephalopod appendages, arms and tentacles, are patterned by a gene regulatory network that is homologous to those active in vertebrate fin/limb buds and arthropod legs. The authors make their case through examination of the RNA expression patterns of key members of the appendage gene regulatory network as well as experimental manipulation of the dorso-ventral and the anterior-posterior axis, yielding results consistent with what is known from vertebrate and arthropod appendages. The authors do carefully consider and test alternative interpretations of their findings, and I find the corpus of evidence convincing. These findings support the notion that there is a patterning module that existed in the most recent common ancestor of bilaterians and which has been inherited to "limbed" members of all major bilaterian clades.

As all important discoveries, this too raises new questions that the authors may want to consider more extensively than done in the submitted manuscripts. If, in fact, the limb patterning module is as old as or rather older than the most recent common ancestor of bilaterians this module must have played a developmental role in that ancestor. The authors say that their findings do not imply that this ancestor had "limbs" but then what was it doing? Also, there has to be a function for this module that explains its conservation from the bilaterian ancestor to the modern groups which have, independently, evolved limbs. Is there a non-limb/appendage function known of this patterning module, or do we have to assume a massive loss of appendages in the evolution of the non-limbed crown bilaterians?

*Reviewer #2:*

The authors describe the molecular mechanisms underlying the development of appendages in a cephalopod mollusk. The article focuses on the expression and function of genes involved in limb development known from arthropods and vertebrates and thus presents the first comprehensive analysis of appendage formation in Spiralia.

The authors state convincingly, that the spatio-temporal expression patterns of the genes analysed show conserved aspects to the known determination networks, as well as the function of Bmp and Hh signalling. They also propose a molecular reason for the morphological difference between arms and tentacles.

From their findings the authors conclude, that not the bilaterian limbs are homologous, but rather their underlying developmental mechanisms.

The in situ data are of acceptable quality, clearly presented and the manuscript is very well written.

The authors state, that expression patterns along the proximodistal axis are "similarly polarized" when compared to arthropods and vertebrates. However, especially in the case of *exd* and *hth* the expression patterns vary among arthropods. Similarly, *dac* usually marks the median part of an outgrowing limb in arthropods and is not restricted to the distal part. For the dorsoventral axis specification, the article says that *dpp* is expressed dorsally in arthropods. While this is true for *Drosophila*, expression patterns of *dpp* in other arthropods are strikingly different and not thought to be ancestrally involved in axis formation (see Angelini and Kaufman, 2005, for an overview of expression patterns in arthropods). The existence of these all differences between arthropods, vertebrates and mollusks in expression patterns of the genes analysed should be mentioned and discussed. To this end it would be advantageous to depict a schematic representation of the various expression patterns in the three animal groups to aid readers during the discussion.

Additionally, since the authors come to the conclusion that the developmental mechanisms are homologous between the different animal groups. The Discussion should mention the presence of parts of these mechanisms in non-bilaterian groups. For instance, differential expression of *wnt* in the a-p axis of sponge embryos, or the establishment of the oral-aboral axis in cnidarians.

Finally, in most of the phylogenetic trees shown in the supplementary data there are the sequences for *S. bandensis* and not *S. officinalis*. Please clarify why there are sequences from two different species, when the cloning and further analysis was only carried out in *S. officinalis*.

---

## [Author Response]

The reviewers have discussed the reviews with one another and the Reviewing Editor has drafted this decision to help you prepare a revised submission.1) Please consider extending the Discussion along the following guidelines:a) Consider discussing a possible ancestral function of the conserved patterning module identified in your work, if the original function was not limb patterning (suggested by reviewer #1). Parts of the gene patterning functions are actually found in phylogenetically older animal groups: could these mirror aspects of the original function of the patterning module? (suggested by reviewer #2).

We have added a discussion of the possible ancestral role(s) of the genetic circuit that controls appendage development (Discussion paragraphs three, four and five). We propose that this developmental program might have its roots in the development of non-locomotory appendages (outgrowths from the primary body axis) in other metazoans, including those that were limbless. We also note that these signaling pathways are known to function elsewhere in the bodies of vertebrates, arthropods, and cephalopods. We refer to two previously proposed hypothesis that are consistent with the idea that this genetic circuit originally patterned the anteroposterior axis of the trunk (or the cephalic neuroectoderm) in bilaterians and was repeatedly co-opted to perform the same function (axial patterning) in secondary outgrowths, including locomotory appendages (Discussion paragraph three).

The Discussion also addresses the implications of our finding that these genes are expressed in the developing funnel/siphon of cephalopods (paragraph four). In a new supplemental figure (Figure 5—figure supplement 1), we show that *Wnt5* and *Exd*, which are involved in proximal-distal limb patterning, also are expressed in the funnel/siphon, and the patterns that mirror their expression domains in the limbs. We discuss the possibility that the same genetic circuit could control funnel/siphon and limb development, and that its role in funnel development could potentially predate the origin of cephalopod limbs. Therefore, in the broader context of animal development, independent evolution of appendages could be explained by the parallel redeployment of an ancient developmental genetic program that predates the structures themselves and very likely controlled the development of non-limb outgrowths, such as vertebrate genitalia, beetle horns, or mollusk siphon. Accordingly, application of the term “limbs” to cephalopod arms and tentacles does not make them any more closely related to vertebrate limbs than to vertebrate genitalia or any other animal appendage.

b) Consider including a schematical summary of the gene expression patterns in the cephalopod compared with vertebrates and arthropods, to make it easier for readers to appreciate the similarities between these animal groups (suggested by reviewer #2).

Thank you for this suggestion. We have created a new schematic figure (Figure 5) that compares gene expression in vertebrate, arthropod, and cephalopod limbs. This schematic figure shows the conserved and the variable patterns of gene expression for most of the genes analyzed in our paper compared to published expression patterns in arthropods and vertebrates. The references used for the construction of this figure are cited in the figure legend.

2) In most of the phylogenetic trees shown in the supplementary data there are the sequences for S. bandensis and not S. officinalis. Please clarify why there are sequences from two different species, when the cloning and further analysis was only carried out in S. officinalis. (suggested by reviewer #2).

We have clarified in the main text that while most of the gene cloning was performed in *Sepia officinalis*, some clones were isolated from *Sepia bandensis*, a closely related species. We show that the *S. officinalis* and S*. bandensis* clones have high sequence similarity (Figure 3—figure supplement 1), and comparison of mRNA localization in *S. officinalis* and *S. bandensis* embryos by in situ hybridization showed that the expression patterns in the developing limbs are indistinguishable in the two species. These new data are shown in a new supplementary figure and legend (Figure 3—figure supplement 2).

3) Please also revise the following points (combined from the suggestions by both reviewers):a) You mention in your manuscript that there are no other mollusks that have appendages. Could the captacula of Scaphopods be homologous to cephalopod tentacles? If so, this could be worth mentioning.

We are unaware of evidence supporting structural homology of scaphopod captacula and cephalopod tentacles. It appears that these appendages have different embryonic origins, which would suggest that they are not homologous structures (sensu, derived from a common ancestral appendage). To the larger point, we have added a discussion of polychaete appendages and we propose that the gene regulatory network that we identified in cephalopod arms and tentacles plays a general and conserved role in bilaterian appendage development, irrespective of the type of appendage (Discussion, second paragraph).

b) Discussion: expression patterns of hth, exd, Dll, and dpp can be found for Platynereis during caudal regeneration in Grimmel et al., 2016 (doi: 10.1186/s13227-016-0046-6).

We have added this reference and discussed its implications for parapodia and for the broader question of appendage evolution in Bilateria.

c) Figure 1G: the legend for the DNA counterstaining should be called Hoechst.

Thank you. This has been added to the legend.

d) Figure 2: each panel of the composite should have its own label (same for Figure 3F-I). For better overview, all limb buds should be shown in the same orientation. In some in situ pictures it is quite hard to see the general shape of the limb buds. Here, a dotted outline would make the pictures much clearer (similar to Figure 4H) (same for Figure 3 F-I).

We have labeled the stage on every panel. However, we have retained the orientation of the limbs in order to enable readers to easily see the differences that we describe for cephalopod limbs. This is particularly important for the expression of *Hh* and *Patched*, because orienting them in the same way as the other limb buds could inadvertently lead readers to misinterpret the pattern as dorsoventrally polarized instead of anteriorly polarized. We have included axis labels in every panel as an additional tool to help readers with the interpretation of the spatial domains.